# Methodological Framework for Resilience Assessment of Electricity Infrastructure in Conditions of Slovak Republic

**DOI:** 10.3390/ijerph18168286

**Published:** 2021-08-05

**Authors:** Zdenek Dvorak, Nikola Chovancikova, Jozef Bruk, Martin Hromada

**Affiliations:** 1Department of Technical Sciences and Informatics, Faculty of Security Engineering, University of Zilina, 01026 Zilina, Slovakia; nikola.chovancikova@uniza.sk; 2Institute of Lifelong Learning, University of Zilina, 01026 Zilina, Slovakia; jozef.bruk@uniza.sk; 3Department of Security Engineering, Faculty of Applied Informatics, University of T.Bata, 76005 Zlin, Czech Republic; hromada@utb.cz

**Keywords:** electricity infrastructure, resilience assessment, flood risk, new framework

## Abstract

The quality of the environment as well as public health is convincingly coupled with the functioning of a power subsector. The power subsector plays a pivotal role in the sense that it emerges as the key cross-sectional element for the society’s functioning (production, services, healthcare, education and others). A modern society consists of infrastructure systems that are primarily dependent on continuous electricity supplies. Each and every element of the electric power infrastructure is unique, and thus, its malfunction can disrupt the functioning of an important part of the electric power infrastructure. In conjunction with ensuring the functioning of electric power infrastructure, our attention must be drawn to the resilience issue. As far as the resilience of electric power infrastructure is concerned, it can resist weather-related events ensuring there are no disruptions in continuous electricity supplies. First, in the introductory part, the article presents the legal framework in the Slovak Republic. Second, it describes the current state of the electric power infrastructure of Slovakia. Third, it handles the state of the level of security risk assessment. Later on, in the literature review, besides turning to the issue of resilience assessment, the authors focused on the area of resilience of power engineering. Furthermore, the article scrutinizes resilience assessment in Slovakia, and it briefly examines approaches towards natural threats. In addition, the article demonstrates several approaches towards flood resilience. Having used different methods, the primary concern is to devise a framework for resilience assessment. Therefore, the included case study examines aspects of the proposed framework for resilience assessment. In conclusion, our aim was, in most respects, to outline an innovative methodological framework for increasing the resilience of electricity infrastructure.

## 1. Introduction

The beginnings of theoretical research and development of resilience are associated with the work published by Holling [1] in 1973. The first nuances of the term resilience were contextualized in ecological systems. Accordingly, the concerns about the resilience led to discussion among various scientific disciplines. In line with this, various approaches were engaged to explain as well as to assess resilience. For the purposes of the article, we underscore the definition which approaches resilience as the ability to absorb, adapt and recover, in a timely manner, from the elements’ outage after an adverse event has occurred. In this context, our primary objective, based on elaborated analysis of information resources, is to suggest the methodological framework for the resilience assessment of electric power infrastructure. Developing the methodological framework will, in line with this, draw on the alteration of existing and already available methodologies, e.g., CIERA [2], state energy resilience framework [3] and guidelines for critical infrastructures resilience evaluation [4]. More precisely, the methodological framework will involve five steps, thus allowing effective capture area of resilience and its assessment. Ultimately, the proposed framework shall make a contribution to both education and for practice in Slovak Republic (SR).

The power system emerges as one of the most significant economy segments. Electricity has been produced since 1884 in our country and the development of the electrical power network was gradual, being largely related to the industrial development in Slovakia. The actual backbone of the power network was finished in 1970 [5].

Security and resilience of power networks play a pivotal role in ensuring all functions of modern society. The electricity sector, as a whole, is a crucial element in every citizen’s life, plays a decisive role in the functioning of every business as well as in ensuring all the nation’s functions and, broadly speaking, of the whole world. The more modern the society the greater is its dependency on electricity. Accordingly, objects included in the infrastructure of the power system have been protected for decades under conditions of the SR [6]. The ministry of the Interior of the SR and other central state administration bodies under its control already began to consider the issue of critical infrastructure in 1999. Later on, critical infrastructure appeared in Act. no 319/2002 Coll. on defence of the SR, in which §27 identified the land targets as being of particular significance. At first glance, these land targets played decisive roles in defence. Nevertheless, their significance counted them among the elements of critical infrastructure of utmost significance. A noticeable advance in the field of critical infrastructure is an approval of the national programme for the protection and defence of critical infrastructure of SR in 2008 by means of which 9 sectors were identified. At present, Act no. 45/2011 Coll. on critical infrastructure turns to the subject of critical infrastructure. This law solemnly determines the sectors of critical infrastructure within the area of the SR which are: transport, electronic communication, power sector, postal services, industry, information and communication technology, water and atmosphere, healthcare, finances and agriculture including subsectors. Sectors in the scope of central bodies are defined in Act no. 45/2011, no. 3 in the appendix. Act no. 45/2011 in conjunction with the council’s guideline 2008/114/EC on identification and designation of European critical infrastructures set out to improve their current protection and enhance defence of infrastructures of utmost significance at higher level due to growing terrorist threats. Changes in both inner and outer security environments lead to the release of the new “Security strategy” in response to new threats covering a full range [7,8,9].

The research in the area of critical infrastructure protection has focused on creating systems of both static and dynamic resilience of critical infrastructure. There were defined areas, e.g., terrorism, extremism, organised criminality, influence of foreign power, migration problems, natural risks, anthropological threats, cyber threats, hybrid threats and, also in relation to the topic of the paper, attention must be drawn to the security of the power system. First, researchers set out to carry out thorough analyses, and create scenarios of possible measures in every area of research.

Second, another sort of approach emphasized the research of pillars of the safety management system—physical security, fire security, environmental security, security of business, safety and health protection at work and cyber security. More precisely, several hundred security indicators were defined in the research framework that should provide the basis for intelligent, knowledge-based systems intended for utilisation in the future security management framework [10].

From the writers’ point of view, the paper intends to contribute to the wealth of knowledge focused on flood resilience in weather- and climate-related changes within the electricity sector. The paper, considering thorough analysis of the long-term research topic, sets out to propose methodology oriented towards the resilience enhancement of the whole power sector and, more precisely, to the electricity sector. The methodology was tested on the case study. Additionally, the methodology was verified by a group of experts from academic milieu and in practice. The contribution of the paper will be an outline of measures for increasing the security in the electricity sector.

## 2. A State of the Art

Preliminary work for the scientific paper focusing on weather- and climate-related changes involves a detailed analysis of the resilience issue in the field being explored, a literature review and use of the appropriate scientific methods. Applied research in the area of security is particular in a way that researchers have to consider the legal framework as well as technical norms managing the multidisciplinary domain. The power system is thought to be one of the most complex and significant systems. Due to this fact, the power system represents the backbone and, figuratively speaking, nervous system of modern societies providing basic services including the energy supply, thus ultimately making society’s life easier, increasing productivity, improving business and stimulating economic grow [11].

### 2.1. Juridical Framework in Slovak Republic

“Act no. 251/2012 Coll. on Energy and on Amendments to Certain Acts;Act no. 250/2012 Coll. on regulation in network industries;Act no. 142/2000 Coll. on metrology and on the amendment of certain laws;Act no. 309/2009 Coll. on the promotion of renewable energy sources and high-efficiency cogeneration and on the amendment of certain laws;Decree of the Ministry of Economy of the SR no. 270/2012 Coll. on professional competence for business in the power sector;Decree of the Ministry of Economy of the SR no. 271/2012 Coll., which lays down details on the scope of technical conditions of access and connection to the system and network and the rules of system and network operation;Decree of the Office for Regulation of Network Industries no. 275/2012 Coll., which establishes quality standards for electricity transmission, electricity distribution and electricity supply;Decree of the Ministry of Economy of the SR no. 416/2012 Coll., which lays down details on the procedure for applying restrictive measures in the event of an emergency and on measures aimed at eliminating the emergency situation in the electricity industry and details on the procedure for declaring a crisis situation and its level, declaring restrictive measures in the gas industry for individual categories of gas consumers, on measures aimed to eliminate the crisis situation and on the method of determining restrictive measures in the gas industry and measures aimed at eliminating the crisis situation;Government Regulation no. 426/2010 Coll., which lays down details on the amount of levies from electricity supplied to final customers and the method of its selection for the National Nuclear Fund for the Decommissioning of Nuclear Facilities and for the Management of Spent Nuclear Fuel and Radioactive Waste “[12,13,14,15,16,17,18,19,20].

The acts, decrees and regulations mentioned above play decisive roles to ensure the operation of the power system infrastructure in SR. “Act no. 251/2012 Coll. on Energy” [11]. Accordingly, the first part of the act discusses the essential clauses defining the essential terms in § 2b that are pivotal for guidance in the electricity infrastructure issue. The act addresses running a business in the power sector, it amends the electricity sector, it gives a description of technical equipment and, finally, some other aspects of electricity infrastructure [13].

### 2.2. Electric Power Infrastructure of Slovakia

The electric power grid in most developed countries is one of the largest, the most complex and the most modern systems in the world. The history of the development of the electric power grid in Slovakia went hand in hand with traffic and industrial development along with an increase in citizens’ mobility. The electric power grid was installed, in a variety of ways, in less crowded regions of the country. The development of Slovakia led to the current state, where the Slovak electric power infrastructure includes 37 power plants comprised of 31 hydroelectric, 2 nuclear, 2 thermal and 2 solar power plants. The power plants in Slovakia have 4081 megawatts of installed capacity [21]. The highest share of the production of electric power are nuclear power plants 53.7%, the second largest with a share of 21.7% are thermal power plants 16.1% is generated by hydroelectric power plants and renewable resources generate 8.1% of the total power [22].

For transmission and distribution of the electric power from area of production to the area of consumption is the responsibility of the electric power system comprising transmission and distribution equipment. 

Transmission systems are utilised for the transfer of the high outputs among primary nodes of electric power systems. “Act no. 251/2012 Coll. on Energy” explains transmission as “mutually connected electrical lines primarily of high voltage and very high voltage and electric power equipment needed for the electricity transfer” [13]. A transmission system is made up of lines which are connected by transformers (substations) which, in turn, transfer electricity in as efficient way as possible. The electric power system in Slovakia is operated by the Slovak electric power system (SEPS), which is responsible for the technical condition and flawless operation of the whole transmission system.

Electrical substations are equipment which are situated at selected places on the transmission system for distribution of electric power. The electrical substations feature as nodes in the electric power system. The electric substations include transformers as well as switchgear and control house equipment systems, breakers and protection relays [23].

An implementation of electrical substations into the electric power system determines their significance. Considering this criterion, electrical substations are differentiated as follows:

Transforming stations—their task is to lead the electricity produced into the electric power system and, in turn, transforming stations transform the voltage to the voltage of the network they are connected to,

Electricity network nodes—are also called switching substations, they create a common point among the branches of the circuit transmission lines and nodes distributing the electric power of the same voltage;Transformer substations—they serve to connect transmission systems with different voltages and serve to transform 400/110kV and 220/110kV;Distribution substations—their function is to distribute electric power and to transform very high voltage to high voltage and, moreover, they deliver electric power to the consumer centres;Industrial electric substations—they divide the electric energy in the industrial companies directly to the appliances and they transform high voltage to low voltage [24].

**Distribution of electric power** is the final stage of the electric power supply. The distribution substations transfer electricity from the transmission system to the individual consumers. “Act no. 251/2012 Coll. on Energy” defines distribution systems as “mutual connected lines of very high voltage up to 110 kV including high or low voltage and electric power equipment needed for distribution on defined area” [13]. 

“Act no. 251/2012 Coll. on Energy” (the Act) on energy assumes two kinds of distribution systems—regional and local. The difference between regional and local distribution system depends on the number of connected consumers. The local distribution system enables connection of up to 100,000 consumers, whilst the regional distribution system allows connecting more than 100 consumers. Besides that, the Act also defines an operator. Accordingly, three operators exist providing electricity in the area of the SR: Westslovak distributional Inc., Centralslovak power—Distribution, Inc. and Eastslovak distributional Inc. The companies mentioned above supervise the quality of provided services along with the distribution of electric power [25].

### 2.3. Introduction Analysis of the Security Risks

Many adverse events might endanger such a very complex system like an electric power network. Threats of natural character might jeopardize the area of SR and individual electric power elements, respectively. Ultimately, a clear distinction must be made between threats of natural character, e.g., floods, or threats of anthropogenic nature, e.g., terrorism. Nowadays, the serious changes in climate conditions are attributed to occurrence of weather-related events. Undoubtedly, the influence of weather-related events will more frequently cause disruption in operation processes across the whole area of the SR. More specifically, an example of such a weather-related event in practice was strong winds associated with a hurricane that attacked parts of our country on 24 October 2018.

The hurricane caused power outages and traffic collapse. Just short of 55,000 Slovak households were left without electricity. As far as weather-related adverse events are concerned, heavy snowfall caused problems in many places of the SR in 2019. The adverse events were, in turn, reported on the web portal webnoviny.sk announcing disruptions on electric lines as a result of heavy snowfall [26]. Researchers established that there were no blackouts in the area of the SR during the researched period. On the contrary, cases of severe power outages are provided by Italy, which had a problem with blackouts having impacted the whole peninsula for 12 h, and Switzerland, where some parts were affected for approximately 3 h. On the whole, it was the biggest outage, affecting 56 million citizens, across Europe. With respect to its cause, disruption of electricity supply was triggered by the storms that damaged the transmission line from Switzerland to Italy.

Furthermore, another stress associated with Italy emerged because of disruption of the electricity supplies between France and Italy being caused by abrupt increase in electricity demand. A trigger cause of the blackout was unsatisfactory trimming of bush and tree branches under high voltage transmission lines. In line with this, the outage ultimately resulted in several hundred people being trapped in underground carriages. Considering the issues related to air transport, all flights were cancelled because of the outage [27].

Natural disasters do not seem to be the sole reason possibly influencing the function of the run-in processes. Although anthropogenic threats might be avoided by applying the appropriate preventive measures, security engineers cannot with certainty exclude personnel threats. Thus, for anthropogenic threats, as opposed to natural disasters, human factors are responsible. The actions of man influence various factors, which are reflected in carrying out his job. Specifically, negative factors, like stress or psychological disorders, may give a rise to crashes, sabotage or, in in the worst-case scenario, terrorist attack. According to the available resources, electric power infrastructure in the SR have not been damaged by threats of anthropogenic nature so far. For the most part, short-term outages were caused either by small technical malfunctions or outages occurred having apparently not influenced everyday life.

The case for such blackout abroad happened in Venezuela on 7 March 2019 when literally impacting whole country. There was a traffic jam in the capital and underground service outage occurred. It was the largest outage in the history of the country, because it lasted for 7 days. The electricity outage caused, in terms of loss of life, the death of the 17 patients who were hospitalised. The government considered it a sabotage and accused the USA of possibly having disrupted the key power plant, Guri. As a result, Venezuela’s president accused the USA of carrying out the cyber-attack on the automatic control system, even though experts’ opinions claim that the outage was caused by insufficient investments into infrastructure [27].

## 3. A Literature Review

The literature review is focused on analysis of available resources addressing resilience assessment in general. Furthermore, writers were especially mindful to search for the topics resolving the issues of resilience assessment in the electricity subsector, articles dealing with possible sources posing threats and potential risks posed by floods to the electricity subsector. Needless to say, the highest occurrence of changing disaster risk management related to floods in the SR was our greatest interest.

### 3.1. Analysis of Approaches towards the Resilience Assessment

Assuming that research into the issue of resilience, and its assessment, underscores active approaches towards analysis of large amount of information resources, the writers hope to shed light on actual scholarly-scientific information. The approaches towards resilience assessment are established based on the combination of qualitative and quantitative methods that enable measurement of the resilience level of objects assessed. In line with this, active approach was recently recorded in the area of critical infrastructure assessment (CIA).

In the publication, “Assessing and measuring resilience“, explored the resilience assessment and its measurement listing infrastructure systems affecting everyday life. Firstly, she produces interesting insights into resilience with focus on roads and hospitals, which were impacted by supply shortages due to adverse events, e.g., floods. In her opinion, the resilience system depends on the system’s features. Secondly, for definition, quantification and overall proposal of resilience enhancement, Proag introduced features as follows: absorption, adaptations and renovation. The author also mentions steps for carrying out resilience assessment of social-economical systems. Thirdly, indices within the scope of individual infrastructure are also mentioned, e.g., emergency services—number of saved lives, telecommunication—number of interrupted phone calls, among others. An essential part of the paper is qualitative as well as qualitative resilience assessment. To sum up, she resolves the problem of resilience efficiency, on the one hand, with quantitative assessment. On the other hand, qualitative assessment is carried out by risks analysis, thus revealing risk recourses [28].

The writers of the paper, “Resilience assessment in electricity critical infrastructure from the point of view of converged security“ place great emphasise on the electrical power sector which counts among the most salient critical infrastructure sectors. Provided that the power disruption might largely impact citizens’ lives, economics as well as safety, it is crucial to ensure comprehensive monitoring resilience level of critical infrastructure elements. For the sake of monitoring, the writers developed the Converged Resilience Assessment (CRA) method allowing measurement of elements’ resilience levels in power system infrastructure from the viewpoint of convergence safety. More precisely, the CRA method is coupled with informative and situational management. Both roles of management aim to integrate and correlate the information from the systems, by means of which they get an overview of the situation. This overview, in turn, provides the basis for receiving effective resolving of defined states. Figure 1 depicts the proposed approach in more detail. The approach gives an explanation for a layout of the resilience assessment from the viewpoint of converged security [29].

The scholars of the paper named “A quantitative method for assessing resilience of interdependent infrastructures” gave their attention to resolving the issue of resilience assessment of mutually dependent infrastructures. In line with the assessment framework, their investigation brought encouraging results after applying the proposed methodology. More specifically, the writers put emphasise on the significance of system resilience comprehension identifying the ways of how to achieve resilience improvement. It should be noted that resilience improvement is aimed especially at mutually interdependent infrastructures, which our everyday lives depend on. Their contribution relies on the quantitative methodology outline being used for system resilience assessment. The suggested methodology involves two components: first, integrated metrics for system resilience assessment, and second, hybrid modelling approach for representation of faulty behaviour of infrastructure systems. The investigation results apparently prove effectiveness of the methodology at suggesting and improving infrastructure resilience [30].

For the sake of examination of static resilience assessment, considering the paper, “Resilience measurement index: an indicator of critical infrastructure resilience” (RMI) is needed that was written by a group of authors in the national laboratory in Argonne. When designing RMI, the authors drew on the present demands of the United States of America (USA). On the whole, the 21st century is marked by the formation of many threats, that are of random nature. Threats of different natures might undermine significant functioning of the society that, in turn, affect the wellbeing of society. The writers claim that overall improvement of the resilience of the national economy calls for a comprehensive approach including the need for widening the current approaches. Accordingly, it is unsatisfactory to focus solely on prevention or mitigation that have previously achieved significant results. The writers are aware of the necessity to create the methodology which would comprise protection, as well as readiness, mitigation of impacts, response and recovery. The projection of RMI was designed in order to cover basic resilience-driven features with respect to all risks. Therefore, the main aims of RMI are ability to measure level of critical infrastructure and ability to decrease either the range, long-term impacts, or in combination, of adverse events. The RMI methodology might be coupled with other indices, e.g., index of impacts measurement, and according to the opinion of the writers, it will thus ensure a comprehensive approach in terms of the resilience assessment [31].

“Guidelines for critical infrastructures resilience evaluation” presents a bunch of directives from practice that are supposed to be offered to the specialist in a way they can assess resilience of all critical infrastructures. This document puts forward an assessment of static resilience applied in general in different sorts of infrastructures. The assessment of static resilience constitutes proposed indicators, which can comprehensively cover the assessed sphere. Indicators within the directive’s framework were projected for each individual resilience sphere, e.g., logic, physical and personal. By applying and subsequent assessment of indicators, evaluators can achieve comprehensive resilience assessment levels of a component in critical infrastructure [4].

Within the territory of the Czech Republic, the “Methodic of rating resilience of critic infrastructure component” (CIERA) was created, which was performed with grant support VI2015201949 “RESILIENCE 2015: Dynamic resilience assessment of correlated subsystems of critic infrastructure“ supported by The Department of the Interior of Czech Republic in years 2015–2019. More precisely, the CIERA methodology presents a tool, making critical infrastructure components for operators allowing assessing of levels of static resilience via the suggested algorithm. This methodology might be applied in several sectors. Specifically, the methodology was tested in a variety of ways in the energy sector, in traffic and informational and communication systems. The methodology is a universal tool for resilience assessment, which is capable of resilience assessment within the critical infrastructure, and, furthermore, it makes it feasible to compare resilience among different subsystems of critical infrastructures as well. It is necessary to point out the area of natural hazards both within the CIERA methodology as well as for the purposes of our exploration; accordingly, three groups of natural hazards were identified, that is, climatological, geological and biological threats. Subsequently, threat identification is indexed in assessment sheets which contain both the name and the category of the threat and, at the same time, category defines the robustness of the element. From the methodological point of view, the measurable items determining the robustness are defined in relation to geological, climatological and biological threats. All measurable items are then evaluated at this stage with the respective points. Finally, the whole process calculates the final quantification of the element’s resilience [2].

### 3.2. An Analysis of the Approaches towards the Resilience in Power Engineering

The research being described ultimately narrowed and focused on the issue of the static resilience assessment from the viewpoint of energy infrastructure objects. Accordingly, analysis of the resilience approaches and their assessment from different areas as well as the real background of Slovak power engineering must have been considered. The scholarly article, “A review on resilience assessment of energy systems”, draws our attention to the assessment of power systems’ resilience that form the essential pillars of our society. Functioning of all the processes in society depends on regular supply of electric power. Power systems are impacted by disruptions that, for the most part, impact economic activity, operation of infrastructure subsystems and society in general.

The writers of the article state that the power system is one of the most significant systems, hence, it is necessary to work continuously on enhancing the resilience and simultaneously improving the management of possible impacts of natural disasters, technical malfunctions or accidents caused by anthropogenic agents. The writers carried out the analysis focused on the approaches concerning the resilience assessment of power systems. Pertinent resources were divided according to the approach and, more precisely, they identified four key resilience functions: resistance, renewing stability, recovery and adaptation of the infrastructure. Thus, the research proves that the resilient system always works aiming to minimize potential consequences coming from the adverse event. Ultimately, this system attempts to effectively diminish the potential loss to the output of the system [11].

Sharifi and Yamagata address the theme of energy consumption in city states in their contribution “Principles and criteria for assessing urban energy resilience”. The writers point out that city states and city agglomerations consume 60–80% of the power. The issue of the continuous increase of citizens in the cities deserves a special mention because it goes hand in hand with the increase of the consumption of the electric power in the future. Some crucial elements regarding the continuity comprise climate changes and increasing amount of adverse events, e.g., cyber-attacks, terrorism, technical shortcomings and floods. Ensuring the resilience of the city power system involves fulfilment of conditions as follows: ability to plan and to be prepared, to mitigate threat and to adapt to all adverse events that might occur. An integration of these four abilities into the power system might cover the continuous dealing with availability, accessibility, price convenience and acceptability. The writers of the article chose various plans and constructional criteria categorizing them into seven areas: infrastructure, resources, utilizing the soil, architectural arrangement of city agglomeration, management of public administration, social-demographic aspects and human behaviour. Drawing on the assumptions behind the framework mentioned above, we are convinced of the complexity and multilateral nature of power resilience. Finally, research into the significance of identified criteria revealed the advantages of mitigation and adaptation that a great number of criteria can bring [32].

Writers of the article, “Metrics for energy resilience”, respond to the possibilities of electric power supply disruption by outlining metrics for energy resilience. The energy subsector considers resilience as a significant subsystem crucial for the functioning of other subsystems existing in a country. The resilience constitutes a new emerging concept providing dynamic background uniformly with legacy static measurements and, at the same time, considering the link between changing disaster risk and difficult interactions being found among physical, informational and human areas.

Their article aims to provide a description of metrics to be implemented for the areas of planning, design, investments and operation within the power sector. The scholars outline a matrix that depicts complex network of metrics applied to energy resilience of the system in a tabular form [33].

The article, “Quantitative model and metrics of electrical grids resilience evaluated at a power distribution level“, aims to introduce the framework for systematic measure and assessment of resilience of the power networks. The scholars focus on the transmitted output that customers perceive of the level of energy distribution. The framework being proposed allows measuring the degree of functional dependence of the power network’s load and it points out the nexus of resilience and dependence concepts. The qualitative nature of the framework allows creating the tools for evaluation of the power network’s output as the saving line. Moreover, it enables the effects’ evaluation of the plans for optimal setting as well as operation of the infrastructural networks [34].

“Measuring the resilience of energy distribution systems “is a report which discusses the issue of resilience in general by improving the resilience of electric power infrastructure. The making of the report was supported by US ministry of energy and Bureau for energy policy and analyses of USA systems. Firstly, report writers focus on the summarization of concepts that are part of resilience measurements. Secondly, the framework for the organization of alternative metrics used on measurement of resilience power distribution systems is described and, finally, the document explores the state-of-the-art resilience metrics of these systems [35].

Another useful source related to the topic mentioned above is a scientific article, “Criteria risk analysis of facilities for electricity generation and transmission” [36], and an article focused on the application of the AHP method entitled “Preference Risk Assessment of Electric Power Critical Infrastructure” [37].

### 3.3. An Analysis of Approaches towards Resilience Assessment in Slovakia

An analysis of information resources that are available to the public by means of textbooks or Internet resources suggests that the issue of resilience and its assessment were not comprehensively discussed on national level with an exception of the writers.

Besides that, publications exploring the issue of resilience focus on the resilience of the materials or mechanical design. The research subject, i.e., critical infrastructure resilience (CIR), is addressed by researchers of the faculty of Security Engineering at Zilina University in Zilina. The researchers worked on the topic of resilience publishing several papers. To mention a few, “Modelling resilience of the transport critical infrastructure using influence diagrams“ addresses the impacts of adverse events on the functionality of elements being included among the elements of critical transport infrastructure. The article aimed at the creation of an appropriate approach towards resilience measurement. Hence, coupling of both the theoretic-decisive approach and breakdown time of disruption duration of transport infrastructure elements proved to be an appropriate approach. On the one hand, under the proposed approach, the assessment of the resilience of the elements was conducted. On the other hand, the proposed approach is valid, for the most part, when assessing individual critical infrastructure elements rather than the whole transport network [38].

Secondly, “Critical infrastructure and integrated protection“ is, inter alia, a monograph focused on integrated security system, as it describes mechanical barriers, alarm systems and physical protections in more detail. The writers drew on analysis of critical infrastructure protection [39].

Thirdly, “Resiliency and critical infrastructure protection“ is an insight into the issue of research of critical infrastructure and societal resilience enhancement against natural disasters and disasters caused by man. The research was conducted under the EU Horizon 2020 program. The writer points out great significance in the increasing of society’s resilience against threats of natural character and anthropogenic disasters striving to achieve a decrease in disaster impact on the nation in general. The more resilient the society, the smaller are impacts of adverse events on it. In the writer’s opinion, not one strategy exists that would solely resolve the whole issue of infrastructure resilience, because every system has its own aims, priorities and sources. In most respects, ensuring a sufficient level of resilience for infrastructure systems demands compromises [40].

### 3.4. Analysis of Approaches towards the Natural Threats

The writer of the paper, “Weather-related power outages and electric system resiliency“, scrutinises the electrical power outages caused by the storms. The writer draws on percentages of the weather-related power outages, but, unfortunately, the flood category is not included. The nearest category to floods is wind/cumulative rain. Interestingly, the weather-related event mentioned above causes just short of 15% of electric power outages. Accordingly, measurement of the electric power outages is based on so-called system average interruption duration indices and a system of average interruption frequency indices. In addition, the paper responds to the economic harm caused by storm-related power outages. Finally, the writer claims that if the area is seasonally prone to seasonal extreme weather conditions, it is inevitable that both government and regional government regulation authorities search for ways to reduce the vulnerability of power and transmission systems [41].

The writers of the paper, “Climate change, disasters and electricity generation“, focus on the electricity supply disruptions that are impacted by climate changes [42].

In a similar vein, hydropower plants in power systems of Slovakia are threatened by two risks, i.e., droughts and high temperature. A reason for the occurrence of these weather-related events relies on a lack of water. As opposed to droughts, heavy precipitation might cause floods more frequently. Climate change impacts, as well as changing disaster risks caused by floods, repeatedly resulted in high water levels. Due to this fact, flood gates are opened and, in turn, a great deal of electricity is generated. Nevertheless, another weather-related event, like storms (cyclones), might result in damage of power plant equipment, of the water reservoirs and dam threat.

Climate change and changing disaster risk might impose significant adverse consequences on energy politics and planning. Therefore, possible solutions prove to be the designing of more robust infrastructure and the establishment of disaster risk management. In terms of protection against floods, preferred power plant location and their equipment should be outside the area prone to floods. Besides, the projected sea level or construction of more robust power plant and dams speak in favour of mindful power plant location. As mentioned above, researching the impact of climate changes and disasters in the electricity sector plays a decisive role. More specifically, it is vital to outline adaptation strategies that strengthen infrastructure’s resilience to climate change, increase energy security and, finally, enhance the resilience of low carbon climate.

The paper, “The Vulnerability of the Power Sector to Climate Variability and Change: Evidence from Indonesia“, discusses development of Indonesian infrastructure that is still vulnerable to the natural risks or other causes like slow technology diffusion. The writers of the publication conducted fieldwork, and as such, the paper contains valuable information. The writers recorded all the constituent elements of the Indonesian power sector on Java and Bali islands. Apart from the mapping of the power grid, the fieldwork provides us with description of weather and climate, financial losses suffered and corresponding adaptive responses of the power sector to adverse events. To sum up, the fieldwork proved that the impact of weather and climate, to a great extent, affects the production, transmission and distribution in power sector (Table 1).

The adaptive responses having been made so far are conducted solely on decentralized, autonomous and individual level, not seldom being attributed to the outages having occurred. The planned and aimed strategy that would manage the decentralized adaptation measures is, on the basis of implications discussed, missing [43].

### 3.5. Analysis of the Approaches towards the Flood’s Resilience

The subject of flood risk assessment frequently deserves great attention. National meteorological and hydrological institutes assess daily flood risk and regularly issue alerts of increased flood degrees. Based on recent resources, significant writers (European court of auditors, 2018) released a special report titled “Floods directive: progress in assessing risks, while planning and implementation need to improve “. The writers of the special report put emphasis on the ongoing issue with their negative impacts for the future, e.g., lack of up-to-date knowledge, estimated impact of climate change on the incidence of floods and usage of obsolete data, which, in turn, carry the risk of not reflecting heightened climate risks. As a result, states tend to opt for private flood insurance, where coverage remains rather low. The conclusive recommendations play a key role because they propose to improve accountability, financing of measures focused on decrease of flood risks and ultimately, better integration the effects of climate change into flood risk management [44].

The problem of natural hazards is clearly attracting attention in the sense that there was a paper published within the framework called Economic and Environmental activities of OSCE dealing with electricity networks. Firstly, the writers discussed the rules of disaster risk reduction. Secondly, they dealt with case studies—blackout of 2003 in Italy, Switzerland and Sweden and Denmark, a 2006 blackout in Germany, Slovenia icing in 2014, south-eastern Europe floods in 2014 and storms since 1999 in France. Finally, the most influential part was good practices from private and public sector stakeholders [45]. The study, “Power grid recovery after natural hazard impact “, addresses the topic of critical infrastructure and, in addition, it deals with an estimated impact of electricity supply disruptions being caused by a disaster on society. The disasters constitute, on the whole, danger for all ongoing processes in society and, disasters have a serious impact on citizens’ lives. As far as past experiences are concerned, the disasters proved significant ratios on the critical infrastructure damage and on the adverse consequences caused on society resilience as well.

Firstly, the study describes some energy infrastructure that might be imposed by various threats. Concerning this fact, writers consider energy infrastructure very significant in terms of both society and population. Secondly, the study illustrates the impacts of weather-related events on energy infrastructure also addressing floods. Interestingly, the electric energy supply was disrupted in 100% of flood-related events that were explored in the case study. Furthermore, the case study dealt with the time-based level damage, focusing on when the equipment was inundated. Thirdly, the writers dealt with other aspects influencing recovery time. Within the scope of this section, the writers went into the resilience of energy infrastructure and disruption of other infrastructures. Finally, the conclusion contains many recommendations advancing the progress of the researched topic [46].

The paper, “Critical infrastructure impact assessment due to flood exposure“, discusses the flood inundation impact on critical infrastructure. Floods are considered the greatest risk for infrastructure in Great Britain. The paper contains a case study dealing with damage on infrastructure caused by floods within the Thames catchment area. Floods might result in power outages in flooded areas, thus impacting the surrounding areas that are not directly inundated by floods. This outage would be caused since elements of energy infrastructure are mutually interconnected. The article presents intense debate in the sense that preventive measures should prioritise resilience strategies avoiding the disruptions of the electricity supply for the whole network of dependent infrastructure assets. The presented methodology aims to quantify the number of customers potentially subjected to flood hazard within the Thames catchment area. In turn, the number of potentially flood-affected customers is quantified. In line with quantification of the likelihood of flood exposure, three flood risk levels were defined due to potential disruption. If the coefficient 1 defines the high-level risk, then 1.8 times more customers are at medium risk and 2.5 times more customers are at low risk of electricity disruption. Interestingly, water towers would be impacted indirectly, because their buffer can balance the demand due to gravitation. Although the airports in the Thames catchment area are at the lowest risk and threats are rather indirect, potential outages of Heathrow and Gatwick airports) might have wider social and economic consequences [47].

The paper, “Climate change and critical infrastructure—floods“, puts emphasis on the climate changes causing the high occurrence as well as the severity of hydro-meteorological risks. From the writers’ point of view, adjusting to climate change relies on climate infrastructure resilience. In this context, the scholars carried out a case study in a big western Europe city centre focusing on the power network. As far as the floods in Europe are concerned, 30 out of 34 countries assessed the floods as whole-country related risks. Besides that, 10 out of 34 countries stated that floods might disrupt their critical infrastructure. The damage to equipment caused by floods would be catastrophic if inundated assets were under voltage. The repair of inundated substations is more demanding than repair of fallen power lines occurring even more frequently. The writers claim that a pre-emptive shut-down strategy for inundated substations is effective even if the strategy will cause outage to uninundated areas. As a rule, applying a pre-emptive shut-down strategy eliminates damages to equipment under voltage and shortens the repair time. The case study presents various models that provide a variety of estimations. Thus, one of the most valuable findings in the paper is an estimation that flood-related power outages have 6–8 times higher impact on business than the flood itself. In broad terms, an option for substation protection is their locating out of the area of potential flood levels. To sum up, the paper, for the most part, aimed to demonstrate the assumptions behind critical infrastructure’s risk assessment imposed by floods across Europe [48].

Typically, the most important drawback of literature review is thorough investigation into the resources available. In turn, 26 papers read were used in the contribution presented. Thus, the overview should have structured the topic discussed as follows: Analysis of approaches towards the resilience assessment (7 papers), analysis of approaches towards resilience assessment of the power sector (10 papers), analysis of approaches towards resilience and resilience assessment in Slovakia (4 papers), analysis of approaches towards the threats of natural sources (3 papers) and analysis of approaches towards the flood risks assessment. The writers used ideas, information and knowledge coming from global proposals and they adopted them considering conditions in the SR. In line with this, the methodological framework proposed in the context of the SR is new and original.

## 4. Research Methods

Firstly, the collection of data, along with its collation, contributed essentially to the knowledge base. Secondly, information resources arose out of the energy companies’ data, from the published research reports and the scientific papers. Thirdly, a great number of brainstorming sessions appeared to be another method having been applied. With respect to the current epidemiological situation, writers conducted brainstorming online. On the one hand, online brainstorming improved the cooperation of research workers, and besides, it saved costs needed for research. Furthermore, another benefit was more effective usage of research equipment. On the other hand, online cooperation brought a noticeable level of exhaustion as it was the sole means of handling the work duties. Scholars then studied scientific papers in more detail and organized controlled discussions over the knowledge database gained to choose the appropriate methodologies and examples of good practices in the process of their own research. Spreading of knowledge among researchers and scholars in practice appeared to be beneficial in most respects. The original idea of the methodological framework for resilience assessment of electricity infrastructure arose especially out of consecutive iterations within the research group. A salient contribution proposed the engagement of a co-author from the Czech Republic having added new point of view at designing the framework. At the time there was no pandemic situation, workshop and seminars having taken place proved to be good practice due to the fact that researchers met final users and projects sponsors. Finally, methods databases, approaches, techniques and methodologies have appeared in the field of safety research during the last decade. Database of specific methodologies coupled with exemplar instances by virtue of case studies with assumptions behind security, protections, vulnerability and resilience contributed to the national know-how. Research results always depend on active engagement in the preparation and solution of national and international research projects. In the article, the scholars thoroughly present the results of the past decade focusing on relevant research projects, their own publications and a case study, which is a new and original contribution to the literature.

### 4.1. Research Projects

When evaluating the results of the research, it is necessary to draw attention to all activities in which the researchers participated. In terms of time, the first project completed within the years 2011–2014 was the joint project APVV - “Critical infrastructure protection in the transportation sector”. The project was solved as the first basic research project in the SR and it focused on the issue of critical infrastructure (CI). Conclusive findings wrote up participants on the project website. More specifically, the final results include studies “Assessment of security environment in relation to critical infrastructure protection” and “Public administration competences in protection of CI in the transportation sector”. Besides, there were devised models as follows—general model of risk management in critical infrastructure protection, model for objective risk management of the critical transport infrastructure (CTI) elements, model of rescue services activities in CTI critical points and model for solving economic impacts of possible losses. Project participants thought up a comprehensive methodology focused on object protection of the critical infrastructural elements within the project. In broad terms, the authors of the project made up a unique knowledge base for assessing the risks of infrastructure systems, which is used by a large professional community in SR [49].

Secondly, another performed in the years 2014–2017 was a research project solved within the 7FP—RAIN, which dealt with the extreme weather events on transport and energy infrastructure. On the basis of the project, researchers studying meteorological phenomena, namely, the European Severe Storms Laboratory, The Freie Universität Berlin and The Finnish Meteorological Institute examined the adverse events’ effects on energy infrastructure. Within the solved project, extreme weather events (EWE) across Europe were investigated in detail. The research relied on thorough analysis of extreme climate events in the past: windstorms, heavy rainfall and flash floods, landslides, river floods, thunderstorms, tornadoes, hails, lightings, snow and snowstorms, freezing rains, wildfires and coastal floods. Accordingly, project participants analysed each EWE considering possible impacts on individual sectors. Subsequently, researchers identified possible impacts on objects in the respective sectors and, in turn, they put forward best practices from preventive and response measures. A salient part of the solution is to be found in contribution 2.3 focusing on risk monitoring and warning systems. Suffice it to say that the result of assessment of the state-of-the-art early warning systems and recommendations applicable in European countries was published in the project proceedings. The researchers dealt in a variety of ways with probabilistic evaluation of present and future meteorological and hydrological hazards in Europe. Trends and spatial distribution of EWE risk in Europe were established as the key outcomes of the project [50].

Researchers from the AIA Group carried out the investigation into the impacts on energy infrastructure. The “RAIN Project—D4.2 Protection of Elements and Methods in Energy and Telecom Infrastructure” addresses the protection of elements of energy infrastructure. The protection is based on three security models—proactive, preventive and reactive. A pivotal role in the publication takes on the chapter writing of protective measures in the electricity infrastructure, which are divided into parts, namely:Prevention before the event;Mitigation of the consequences during the event;Post-event mitigation.

In this context, the authors focus on the electricity network and the most common measures applied to the electricity network against meteorological causes. Just to mention a few, among the usual ways of protecting transformers are walls and covers or placement of transformers on elevated levels.

One of the most salient measures for generators is their winterization. With line redundancy, the loss of functionality of the substation (node) is a much more serious case than the rupture of the redundant line. Nevertheless, the failure of generators has even more serious impacts on the entire energy network. Finally, extremely low temperatures apparently have a negative impact on gas and fuel power plants, which in one case (“February 2011 Southwest Cold Weather Event” in the USA) had to be subjected to a winterization process for reliable operation [50].

Another project with our participation conducted within the years 2015–2019 was “Dynamic assessment of the resilience of related subsystems of critical infrastructure”, and it was formed within the security research of the Czech Republic. A large group of authors participated in the elaboration and respective parts were led by Martin Hromada, David Rehak, Pavel Fuchs, Tomas Apeltauer, Petr Hruza, Michal Bil and Vit Stritecky. Firstly, the project set out to scrutinise the static resilience assessment and, secondly, through the prism of dynamic resilience assessment participants attempted to define the possibilities within selected sectors (i.e., energy, transport, information and communication technologies) and their elements. The project provides a description of the synergistic effect of the failure of the mentioned systems and their effect on the impact assessment and the determination of the dynamic assessment of CIR. The empirical part of the project aimed at creating a system for determining the key elements of CI of land transport, CI of the energy and ICT sector in the context of their correlation and in the nexus to the crisis preparedness of territorial units [2].

### 4.2. Design of a Framework for Resilience Assessment

The framework proposed below is a modification of already existing methodologies applied abroad.

More specifically, CIERA methodology [2], state energy resilience framework [36] and guidelines for critical infrastructure resilience evaluation [30] depicted in the analytical part were modified into Figure 2 for the needs in the SR. The procedure depicted in Figure 2 puts forward a comprehensive tool for defining the resilience level. The depicted content offers a somewhat different view of resilience in comparison with the previous documents. Accordingly, the individual features are briefly commented on for the readers. Thus, both components as well as mechanisms of handling the framework are apparent.

The methodological framework is set out in five basic steps, the aim of which is to increase the resilience of the electricity infrastructure. Accordingly, the first step towards resilience enhancement is “defining the system”. Therefore, the writers argue for detailed characteristics of the assessment system. If attention is focused on the electricity infrastructure component in this step, one of the attributes will be, e.g., transmission and distribution system. Subsequently, a detailed characterization of the primary components will be conducted, thus followed by another part of the framework. Secondly, the step “risk identification” establishes risks that might jeopardize the functioning of the components of a defined system. Thirdly, the step named “develop a risk assessment and resilience enhancement process” attends to the assessment of the identified risk from the previous phase and it is comprised of the resilience enhancement process. Next, the risk assessment is performed according to the ISO 31,000 standard, which is an effective tool used in various countries. Subsequently, the assessor proposes solutions leading to resilience improvement in the most threatened parts of the building. Fourthly, the step, “implement the proposed resilience solutions”, involves the implementation of the proposed solutions enhancing the resilience level. A rationale for this step is “prevention, absorption, recovery and adaptation”, which, according to the CIERA methodology developed in the Czech Republic, constitutes a cycle of resilience of critical infrastructure. Within these parts of the cycle, proposed solutions would be adopted to enhance the resilience of the defined system. Lastly, the step, “monitoring”, involves regular system monitoring and maintenance. Applying of the steps shown in Figure 3 can result in the increase of resilience level of the electricity infrastructure. More precisely, resilience improvement will make it feasible to meet the identified goal of “providing a functioning electrical infrastructure so that it is electricity energy available for the need of the state, companies, and citizens”. In the course of our research, the authors concluded that we need to differentiate between natural and anthropogenic threats. Therefore, they continue to address natural threats focusing on floods.

The results of projects mentioned in Section 3.1 were always summarized within scientific papers. As to the first project, publications mentioned above were written up primarily in the Slovak language in order to build a broad knowledge base for experts in practice in their mother tongue. The second international project aimed to present the results of research mainly in workshops, seminars and conferences and for presenting the results in social networks. The third project contributed, to a great extent, to the elaboration of several papers published in known scientific journals and at major international scientific conferences. One of the papers is the article “Self-assessment in the electricity sub-sector”, where the author argues for unique status of the electricity subsector in the PPD-21 document valid since 2013. Accordingly, an increase in the safety of the elements included in this subsector can be achieved by enhancement of resilience level. Likewise, an increase in resistance level of individual elements can be achieved by a correct assessment of its components, i.e., robustness, recoverability and adaptability [51]. In the article, “Research of safety management indicators”, the authors went into the research of safety pillars (physical security, fire safety, safety and health at work, environmental safety and security, operational safety and security and information security) with a focus on safety indicators. The writers came to the conclusion that targeted development of a comprehensive safety assessment system is a multi-level and multi-union task [52].

## 5. Case Study—Testing the Proposed Framework for Resilience Assessment

The 21th century characterizes a rise of threats of both naturogenic and anthropogenic nature. On the one hand, the climate changes result increasingly in frequent occurrence of natural threats, and on the other hand, technological advances and automation processes stimulate anthropogenic threats. Our case study will focus primarily on a sole threat selected from the possible scenario threats. A distinction of several threats including the possible scenario threats, which previously created a base for the ideas within the framework, are beyond the scope of this paper. Therefore, the case study will be focused on the floods that pose risks for the power sector in SR on the account of their frequency.

The case study describes an empirical demonstration of the proposed framework for electricity resilience. The framework will be applied to the distribution system, which is a lengthy element of the electricity system of the SR. The purpose of the case study lies in the demonstration of a new approach, which draws on the knowledge of the projects mentioned in the literature review above. Following these findings, a framework of energy resilience was created, which in our opinion puts forward a new and comprehensive tool to increase the efficiency of the evaluated object. Please note that the proposed approach can be applied in different areas, not solely for electricity infrastructure.

### 5.1. Defining the System

Electricity infrastructure is a set of elements that constitute interconnected grids. The whole electricity subsystem consists of components, i.e., electricity generation, transmission, transformation and distribution. For this reason, functioning links are essential prerequisites for electricity infrastructure. When damage occurs to the assets being integrated into the electricity infrastructure, electricity transmission might be exposed to failure and, thus, potentially causes disruption of the processes in society that cannot maintain its economic and social well-being without a reliable electric power supply. The electricity infrastructure is considered to be a very complex system. Therefore, it is necessary to define the area to be covered by the case study. After analysing the individual components, writers stated that the distribution network counts among very vulnerable components of electricity supply chain, which, at the same time, is the rationale for choosing the theme for our research.

The distribution system is defined in (Act 251/2012) as interconnected power lines of very high voltage up to and including 110 kV and high voltage transmission networks or low voltage distribution networks and power equipment necessary for the distribution of electricity in parts of the defined area; the distribution system also includes metering, protection, control, security, information, and telecommunication equipment and electronic communication networks necessary for the operation of the distribution system; the distribution system typically includes power lines and electrical equipment (or above mentioned assets) which ensure the transmission of electricity from part of the territory of the European Union or part of the territory of third party countries to the defined territory or part of the defined territory if such power line or electrical equipment does not connect the transmission system to the transmission system of third party countries [13].

More precisely, the entire distribution system consists of interconnected power lines of very high voltage up to 110 kV, the high voltage from 22 to 35 kV or low voltage 0.4 kV (400 volts), and, last but not least, power equipment (e.g., transformers) needed to distribute electricity to parts of the defined area. The whole process of electricity distribution can be affected by various extraordinary events, which can interrupt the distribution of electricity to final consumers. The final consumers are for the most part households and small businesses.

### 5.2. Risk Identification

The electricity distribution can be affected by naturogenic or anthropogenic risks. In the case study, we draw attention to naturogenic risks. The current 21st century is marked by a significant climate change, which frequently leads to an increase in risks of the natural environment. Climate change, as a rule, reflects occurrence of high air temperatures, heat and drought, floods and rising sea levels.

Following the mentioned assumptions, our attention will be attracted to the floods, which frequently occur in the SR. The floods pose a security threat that can affect the functioning of the electricity infrastructure, which vital processes in society depend on. Firstly, the risk of floods is especially high in the event of extreme precipitation or prolonged heavy precipitation that, in turn, causes increased water runoff. Secondly, subsequent melting of snow results in rapid increase of water levels (see Figure 3) [53].

Protection against floods in the SR addresses Act no. 7/2010 Coll. on flood protection. The mentioned Act comprises basic descriptions of terms related to the issue of floods. The issue of floods deserves detailed exploration into the levels of flood activity, which involve the degree of flood risk related to the determined water levels, or flows on watercourses and on water structures in flood situations. The flood plans identify three basic levels of flood activity, namely:Degree—state of vigilance,Degree—state of emergency,3Degree—state of threat [55].

Accordingly, Figure 4 depicts the frequency of floods in various stages over the last five years. Based on the statistical data analysis, we concluded that in 2019, the Slovak Hydrometeorological Institute recorded an increased incidence of floods of the degrees 1 and 3 compared to 2018. The floods of degree 2 reached the same number as in 2018. The increased number of floods in Slovakia may be caused by climate change conditions, which were reflected in a higher frequency of extreme precipitation and or as a result of heavy snowfall.

As far as real flooding events of infrastructure assets and equipment damage are concerned, the article attempts to emphasize the importance of addressing this issue as follows. As mentioned, climate change also has a significant impact on more frequent occurrence of floods, which pose a material threat to the electricity infrastructure and its selected elements. Engineers from the United Kingdom addressed the issue of floods and their impact on the functioning of electricity infrastructure. They concluded that power outages caused by floods pose a major threat to the population, as British electricity and communications networks are not adequately protected against this natural threat [56]. The analysis of information resources in the following section confirms our claims and, in a variety of ways, points out the potential negative impact of floods on electricity infrastructure stated by engineers from the United Kingdom. We will use examples of floods from other European countries and Indonesia to support the argument.

The case for significant damage to infrastructures are the floods in Serbia, Bosnia and Herzegovina and Croatia in 2014. The massive floods caused landslides that disrupted underground infrastructure, transformer stations, and the connection of customers to the grid [57]. As a result of the floods on 25 February 2020, the electricity supply to Jakarta was interrupted. Electricity was interrupted in flooded areas for safety reasons. On 18 November 2019, Austria faced heavy rains, which resulted in floods and landslides. The floods and landslides led to disruptions in rail transport and electricity supply [58]. The events demonstrated the adverse impact of floods on the functioning of the electricity infrastructure and wellbeing of the society. Similar cases put emphasis on the inevitable need to focus on the possible effects of floods and to propose measures that could mitigate the identified impacts on society and damage to the electricity infrastructure.

### 5.3. Risk Assessment Methodology Development and a Resilience Process

The risk assessment is performed through the application of the ISO 31,000 standard, which is the basic tool for risk management. ISO 31,000 is a risk management process that includes precisely defined steps such as identifying, analysing, evaluating and subsequently eliminating risks. The risk management can also be linked to resilience issues. If we want to increase the level of resilience of the object assessed, it is necessary to identify the risks causing resilience reduction and, thus, to make the object more vulnerable. Hence, the following section should to some extent address the process of risk management along with the subsequent interconnection of the resilience enhancement process. In terms of range, this process will not be mentioned in full. Solely, as a characteristic of flood risk, will be part of Section 6.3 [59].

Floods pose the greatest danger to the distribution system. Floods might affect the whole process of electricity distribution, which, as a result, can interrupt the distribution of electricity to final consumers. More precisely, floods can damage the foundations of an electric pole due to erosion or landslides. The most endangered are distribution assets located near watercourses. These elements are characterized by flood activity and can be damaged in the case of 50, 100 or 1000 years of water, respectively.

Figure 5a shows the orientation legend of Figure 5b. Figure 5b shows a section of the flood risk map from the area of the Cadca-district, the municipality of Radôstka village. The flood map depicts areas prone to inundation by 5, 100 and 1000-year-old water that may occur in the defined area. The figure depicts a drawing of civic amenities and areas for housing [60]. During the potential flood scenario, 51 inhabitants of the village of Radôstka would be directly endangered. The floods can disrupt the distribution network, which is made up of electric transmission lines being located near the floodplain. The function of the transformer, which is marked with a black circle on the map can also be compromised. Accordingly, malfunction of the transformer would require decommissioning of the transformer thus resulting in limited supply of electricity to households and civic amenities. Fortunately, there are no businesses in the area that would be threatened by a power outage. After considering the origin of the risk and activating the process of increasing resilience, it is necessary to implement the proposal for a directive to enhance the resilience of the distribution system.

### 5.4. Implementation of the Proposed Resilience Solutions—Prevention/Absorption/Recovery/Adaptation

The implementation of the proposed solutions to increase the resilience of the defined system should be applied in four steps, namely:PreventionAbsorptionRecoveryAdaptation

Ad. 1. Prevention phase

The basic pre-emptive measure to prevent damage to the distribution system is the establishment of protection zones. More precisely, the protection zone (Figure 6) represents the space in the immediate vicinity of the system equipment, which is intended to ensure a continuous supply of electricity. Accordingly, Act no. 251/2012 Coll. on energy, in §43, specifies the requisites related to the protection zone in more detail.

As part of the adoption of measures leading to the prevention of power disruption due to floods in the future, it is essential to focus on the replacement of electric poles that would be able to withstand a flood. Therefore, the researchers suggest replacement of electric poles, which are used for the electricity distribution to households. The reason for replacement of the existing original wooden poles (Figure 7a) which cannot withstand adverse weather conditions (Figure 7b).

Ad. 2. Absorption phase

The CIERA methodology defines the absorption phase, a period that is initiated by the effect of an emergency, e.g., flood. The absorption phase is determined by robustness. The robustness is the ability of an element or object to absorb the effects of an emergency so that the service provided is not interrupted. There are two ways how is it possible to perceive the notion of robustness in the introduced system. Firstly, we can account for robustness structurally and by means of the physical resistance of the element or, secondly, in terms of security, that is, as crisis preparedness. For the absorption phase, we propose the creation of four pillars covering robustness from a structural and security point of view (Figure 8). The pillars were created using the CIERA methodology, consisting of the “Guidelines for Critical Infrastructures Resilience Evaluation”, the publication “Indicators as a tool for assessing the pillars of the safety management system” and the climate change web portal. In the next step of determining the robustness, indicators will be identified in each pillar, which, in turn evaluate each pillar, respectively. The assessment results of the pillars will be a comprehensive evaluation of the absorption phase. Provided with the results, the assessors will be able to establish to what extent the restoration of the damaged infrastructure must be carried out.

For each proposed pillar, specific indicators assessing the relevant area were identified. Each of the indicators was assigned a value of 1 (excellent), 2 (good) or 3 (bad). The sum of all values gives a number that indicates the amount of absorption. A minimum of 20 indicators needs to be designated for each area to cover it comprehensively. To clarify the process of processing individual indicators, Table 2 is given, which provides a brief overview of indicators, including their evaluation. The values were chosen at our own discretion because the researchers could not collect the data by real measurement, as this is sensitive information that is not available to the public. The values 1, 2 and 3 were assigned in accordance with Table 3. Finally, the final assessment will be carried out based on Table 4, which includes the characteristics of individual stages.

In line with the pillars proposed, it is possible to evaluate the degree of absorption from which it is possible to estimate the extent of impacts of, for example, natural threats on the object/element/infrastructure/population. The writers assume that the proposed approach might have effective and applicable implications in practice. To sum up, development of the proposed framework was the subject of the other projects.

Ad. 3. Recovery phase

In the context of the recovery process, it is necessary to focus primarily on the population and businesses that could be identified in the area affected by the power outage caused by the floods. The renewal process is also coupled with unit renewal, which consist of material, technical and personnel equipment. The subject-matter units are divided into specialized units, which are produced by energy companies, whilst other units may be administered by the municipal office (in the municipality), the district office (in the district), the regional office (in the region) and in the state administration. Depending on the extent and location of the emergency, resources must be activated to restore the damaged infrastructure/object/element as soon as possible. From the point of view of a solution for streamlining of available resources, we recommend creating, e.g., a list of endangered persons, which is, in our case study maintained at the municipal office. The list would include the concerned persons dependent on the electricity supply and the list must be primarily filled in. Of course, the people’s comments and addresses are fictitious (Table 5).

Ad. 4 Adaptation phase

The adaptation phase is associated with the ability to adapt the system to a recurring emergency in the future. In this step, the transformers should attract our attention on the account of flood risks. To prevent the damage to transformers, it is possible to apply a system that is established in the United States of America. These measures require mounting of transformers on concrete substrates and enclosing them in locked steel cabinets. Subsequently, it is possible to lift the transformer secured that way by means of a crane to higher platforms, which are intended to prevent the penetration of water into the electrical device [63].

For a more effective evaluation of the prevention, absorption, recovery and adaptation phases, we propose the elaboration of checklists. The checklists’ functions should be complementary in the sense of evaluation of the proposed phases. A possible layout of want sheets (see Figure 9) is based on the application of the PDCA cycle, for each of these phases it is necessary to define all four steps of the PDCA cycle and gradually check 4 ∗ 4 activities that should be performed within each phase.

### 5.5. Monitoring

Finally, monitoring is the last step in the proposed framework. The monitoring is used to supervise the situation around the distribution system, which can be performed, for example, by using drones. By means of supervision, one can detect possible damage in advance. A very important aspect in the security process is maintenance, which should be performed regularly to avoid possible failures.

### 5.6. Partial Conclusion of the Case Study

The case study is being conducted on floods, which may currently pose a primary risk to electricity infrastructure. Floods can affect the electricity infrastructure assets, thus causing disruptions of electricity supplies. The empirical section of the case study is devoted to the use of the proposed framework of electricity resilience. The whole framework process was tested on the example of the village, Radôstka. The energy resilience framework represents a new and effective way of how to increase the level of resilience. Due to the pandemic situation, the methodological framework was discussed only in the academic environment and the elected experts agreed on it.

## 6. Discussion

The resolved topic—flood resilience in electricity infrastructure—is a part of safety research, which the authors have been focusing on for a long time. On the one hand, the University of Zilina has had faculties focused on safety and electrical engineering in its structure since its establishment in 1952. Currently, researchers are using top laboratories in the newly built University Science Park and Research Centre. As a whole, the mutual cooperation of researchers within the university is a key competitive advantage. Concerning international cooperation, the Faculty of Applied Informatics has existed within T. Bata University in Zlin for 15 years. Its key subject in the field of security is security and protection of critical infrastructure. The most salient scientific workplace is the CEBIA—Technical Regional Research Centre.

### 6.1. Defining the System

The electricity subsector is a key sector of modern society. Electricity is needed for all private, corporate and society-wide activities. Flawless operation of the power system is on the whole a necessary condition for normal life. The electricity subsector includes the generation, transmission, transformation and distribution of electricity. In the conditions of the SR, private companies with state participation produce and distribute the electricity and the transmission of electricity is carried out by the state joint stock company SEPS (TSO operator) [6,8,19,20]. In addition to the institutional framework, the second essential component is the company’s interest in energy security as one of the primary priorities [12,28,31,34,35,39,46,47]. Accordingly, security research in the world is focused on two parts: safety and security. The two parts create a comprehensive view of security as a whole, which needs to be complemented by individual levels—global, national, regional, corporate and individual. These five levels always require a specific approach depending on:individual needs and perceptions of security;corporate security culture;regional and national specificities;current global problems (in the years 2020 - 2021, i.e., the COVID - 19 pandemic).

### 6.2. Risks Identification

The authors drew their attention to safety and security aspects in technical infrastructures [58]. The introductory part spoke of electricity infrastructure, i.e., practical application and best practices of risk assessment. However, its results are not listed in the article, as it is sensitive information. As far as outputs are concerned a partial output in this area is the knowledge that the age of the building, regularity and quality of maintenance work is key to the safety and security condition of any technical infrastructure.

Nevertheless, creating an up-to-date, comprehensive and intelligent/SMART solution must consider economic and environmental requirements. Current applied security research requires long-term experience, a multi-union approach and a multi-level solution based on a real scientific basis [64].

On the one hand, previously implemented research projects focused rather on technical safety and security of buildings. On the other hand, the current trend tends to include operational safety, physical and information security, safety and health at work, environmental safety and fire safety. The writers are aware of the need to create a theoretical framework step by step, a suitable methodological procedure and a detailed algorithm of selected scenarios, which will become the basis for future IT solutions. In the past, available information and communication technologies did not provide the opportunity to create a comprehensive information system that would use technologies in real time—the Internet of Everything and cloud computing.

### 6.3. A Development of a Risk Assessment and Resilience Enhancement Process

Our knowledge of the risk assessment and resilience process arises in parallel with global direction and goals. That is why we try to transfer ideas from the global, European and regional levels into our own solutions. Firstly, within the United Nations, a global model of sustainable development was developed [65] so that each activity carried out was in line with it. Secondly, promotion of globally defined 17 goals and 197 targets is in line with previous models in the process of being on the right path at the global level. Secondly, an extensive discussion currently exists in the field of critical infrastructure protection, that is within the OECD, focusing on the global topics of digital security risks to energy, transport infrastructure, government and public services [66]. The national legal framework of individual member states implemented the original Directive 114/2008 on the identification and designation of European critical infrastructures in 2011 [67]. Its main task was to introduce a comparable system for the assessment of critical infrastructure elements. A discussion on its amendment began in 2018 resulting in the current draft of the Directive on the resilience of critical entities. Accordingly, convinced of a need for a new directive, they approved a completely new approach to critical infrastructure protection [68]. Directive 12/2020, compared to the original Directive 2008/114/EC, which focused primarily on the transport and energy sectors, draws its attention to the resilience of critical entities. The main objective of Directive 12/2020 is to improve the service provision necessary for the functioning of the national economy as well as the lives of the population by increasing the resilience of the critical entities that, in turn, cover these services. Directive 12/2020 also seeks to ensure a link between the physical and the cyber sector via the application of a legislative framework that includes comprehensive resilience measures.

The actual result of our research is continuous cooperation between the academic and practical spheres. Linking the requirements of practice with the goals of applied research takes on a decisive role. Despite the considerable breadth of cooperation, we focus only on a part of increasing resilience in the article.

### 6.4. Implementing Proposed Solutions and Monitoring

Security research is constantly focused on new challenges and threats. The key is the application of the PDCA cycle with the aim of continuous monitoring and the effort to implement complex solutions. A good example of the direction of research activities is dynamic resilience modelling. The publication “Determinants of dynamic resilience modelling in critical infrastructure elements” writes of resilience is a salient factor in protecting critical infrastructure elements from the undesired effects of adverse events. Accordingly, the higher the level of resilience, the longer the element can withstand the effects of adverse events. The resilience depends on the duration of the adverse event acting on the element. In other words, the longer the adverse event affects the object, the lower the resilience level is. The dynamic resilience allows you to capture changes in resilience at a time when it is affected by negative factors. The authors pay attention to the factors influencing the resilience of CI elements and the nature of adverse events. These factors define the basis for dynamic resilience modelling [69].

The authors of the article bear in mind that security research is a permanent continuous process that must be addressed in multi-sectoral and multi-level ways. Detailed results of security research are often of classified information nature or are considered sensitive information, so it is not possible to publish specific solutions. On the other hand, the generalized results of the methodological framework are a welcome contribution for research and practice in the world. The presented methodological framework for increasing the resilience of the electricity subsector is the original authors’ result.

## 7. Conclusions

The suggested framework towards energy infrastructure resilience assessment of presents a contribution for the SR, because, currently, no single way exists to conduct resilience assessment on local, regional or national level in the SR, compared to abroad. The other countries introduced into practice a variety of approaches addressing resilience issues, which are resolved by virtue of both qualitative and quantitative methods. The resilience issue is in an unexplored field in the SR that scholars have yet to address. Coupling the information from resources abroad and adjusting them for our country’s conditions might bring new and interesting pieces of knowledge. Although the suggested framework takes features of already created methods and approaches, we might state that our framework presents completely new approaches to resilience issues within the SR. Accordingly, particular steps are governed differently and shed new light on of researched fields.

The aim of the article was to propose a new methodological framework for increasing the resilience electricity infrastructure. Writers are aware of the fact that the regionally widespread and long-term effects outage of a power subsector have a huge impact on human health and lives, on the functioning of the corporate environment and society as a whole. Every year, millions of people around the world are exposed to new extreme weather events. These EWEs force whole nations to find new places to live. Viewing Earth globally, the 50 percent limit of urbanization has been exceeded, that is, more than half of humanity currently lives in cities and urban agglomerations with absolute dependence on the functioning of critical infrastructures and especially the electricity infrastructure.

The resilience research proved the tremendous interconnectedness and complexity of systems, their interdependence and considerable vulnerability. The more complicated and intelligent the systems that control the sources of human needs, the greater will be the possible impact on people’s lives and the impact on the functioning of society as a whole. The ever-growing, even extreme, dependence on electricity brings increasing challenges to the protection of the electricity subsector.

Security research leads us to ever new challenges. As part of our research proposals, we have repeatedly encountered the limits of what is feasible. The complexity, interconnectedness and growing vulnerability of vital systems leads to ever new issues and challenges. Furthermore, security research requires coordinated teamwork. If we want to bring real comprehensive solutions, it is necessary to build on all available scientific research information of the research areas concerned and look for synergistic solutions aimed at a new quality of safety as a whole, and, at the same time, solutions based on the new quality of its individual pillars and indicators.

## Figures and Tables

**Figure 1 ijerph-18-08286-f001:**
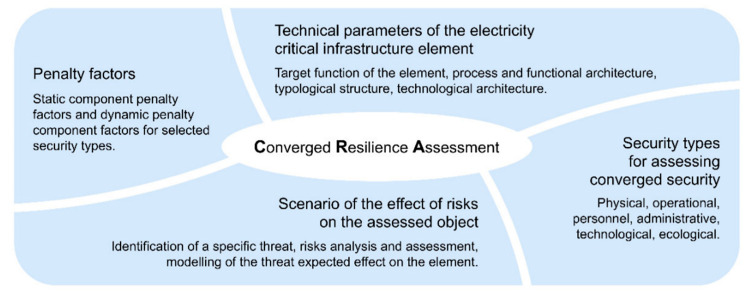
Depiction of Converged Resilience Assessment’s proposal [29].

**Figure 2 ijerph-18-08286-f002:**
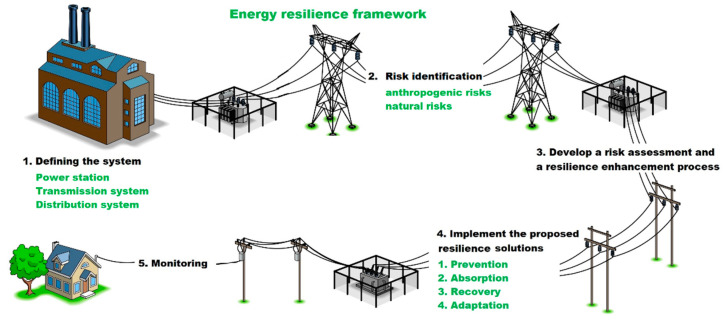
Energy resilience framework adapted according to [3].

**Figure 3 ijerph-18-08286-f003:**
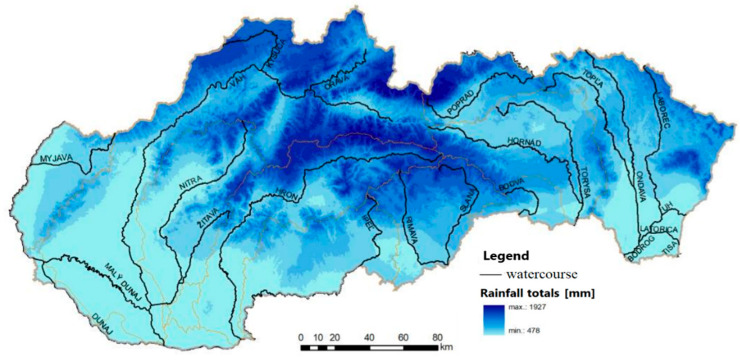
Total atmospheric precipitation (mm) in Slovakia in 2019 [54].

**Figure 4 ijerph-18-08286-f004:**
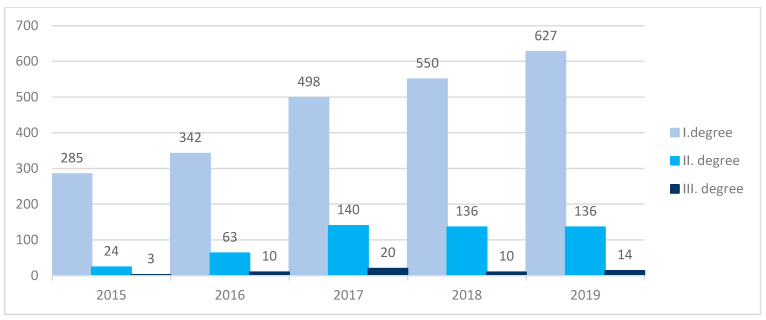
Graphical representation of the occurrence of floods 1., 2. and 3. degree in Slovakia from 2015 to 2019.

**Figure 5 ijerph-18-08286-f005:**
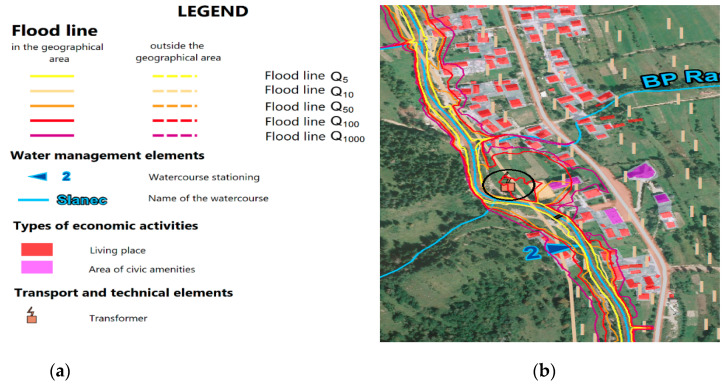
(**a**) Flood map legend; (**b**) flood map of Radôstka [60].

**Figure 6 ijerph-18-08286-f006:**
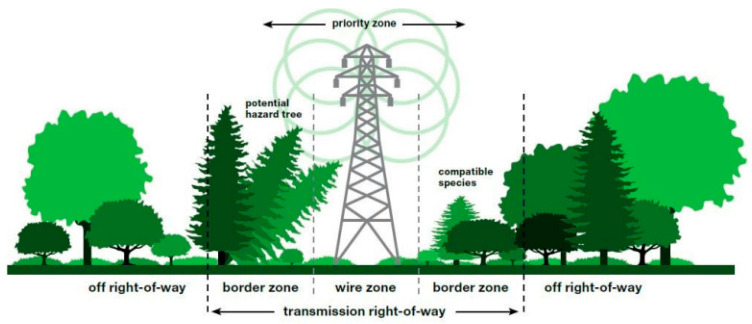
Schematic representation of the protection zone [25].

**Figure 7 ijerph-18-08286-f007:**
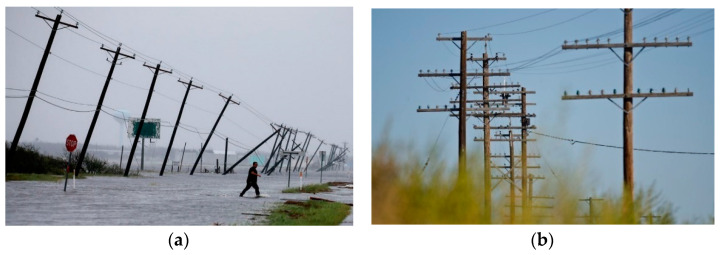
(**a**) Electric pole falls due to floods [61]. (**b**) Wooden electric pole [62].

**Figure 8 ijerph-18-08286-f008:**
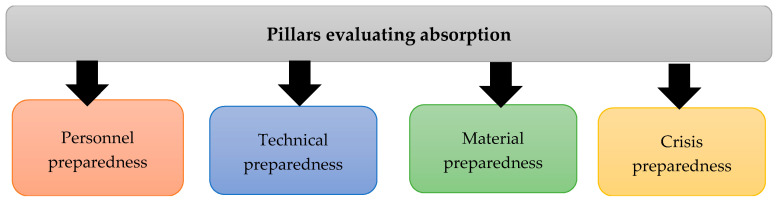
Pillars evaluating absorption.

**Figure 9 ijerph-18-08286-f009:**
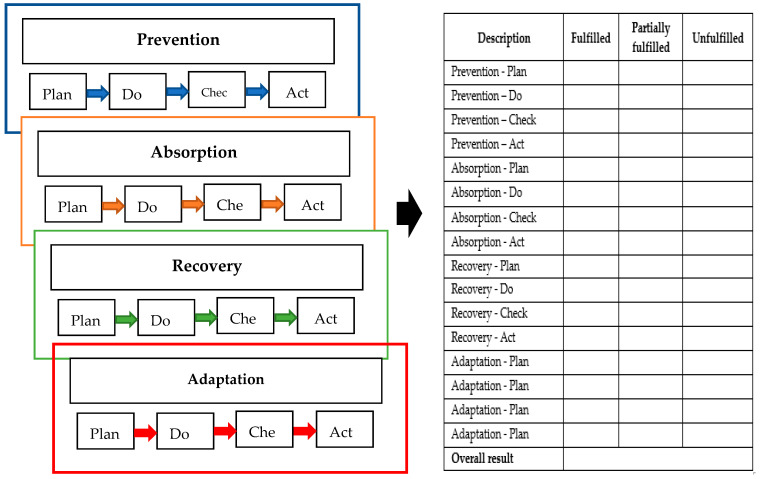
Demonstration of evaluation using checklists.

**Table 1 ijerph-18-08286-t001:** Display of the influence of weather on the production, transmission and distribution of electricity.

Impact Mechanism	Identified Impacts	Adaptation Measures
Days of heavy rainfall caused flooding	Flooded power plant	Construction of flood control systems Mobilization of portable water pumps for draining the water elevation of some areas of power plants
Days of heavy rainfall increased water inflow to the hydropower reservoir, causing the reservoir’s water spills	Flooding downstream	Adjustment of hydropower operation pattern to maintain a normal reservoir’s water level
Sea level rise	Tidal flooding	Elevation of some areas of power plants
Flooded substations	Inundated substations were deliberately turned off for safety reasons and damaged equipment	Elevate the flood-prone substations
Inundated substations were deliberately turned off for safety reasons	Power outages	Identify and elevate flood-prone substations Replace old underground conductors that were flooded Establish a computer application to monitor the area affected by flood-induced power outages Establish a disaster recovery centre and a special service team for post-flood recovery

**Table 2 ijerph-18-08286-t002:** Examples of indicators for individual pillars evaluating absorption.

Pillars	Indicators (I0)	Value		
		1	2	3
		Excellent	Good	Bad
Personnel preparedness	PI1 Personnel maintenance coverage	**1**		
	PI2 Trained personnel working in the field	1		
Technical preparedness	TI1 Flood protection of infrastructure	1		
	TI2 Current technical condition of the infrastructure		2	
Material preparedness	MI1 Number of entities providing material equipment to maintain the functionality of the infrastructure	1		
	MI2 Existence of sufficiently large back-up capacities			3
Crisis preparedness	K1 Analysis of the impacts of natural threats on the built infrastructure	1		
	K2 Providing tools for the effective management of an infrastructure incident		2	
Total number of points		5	4	3
Evaluation of the level of absorption		**12—characterized by high absorption**		

**Table 3 ijerph-18-08286-t003:** Characteristics of point values.

Value	Characteristics
1 (excellent)	Value 1 (excellent) is assigned to an indicator that meets all the specified requirements.
2 (good)	A value of 2 (good) is assigned to an indicator that has gaps within the specified requirements.
3 (bad)	A value of 3 (bad) is assigned to an indicator that does not meet the specified requirements.

**Table 4 ijerph-18-08286-t004:** Determination of the total level of absorption.

Point Evaluation of Absorption	Absorption Characteristics	Degree of Absorption
8–12	The element/object is characterized by a high level of absorption. In practice, this means that all necessary material and physical resources are sufficient so that the object/element can withstand the effects of an emergency and be able to provide basic services until the adverse effects of the emergency disappear.	High absorption
13–18	The element/object is characterized by a good level of absorption. In practice, this means that it is well covered with the necessary materials and physical resources, but with minor shortcomings. The object/element can withstand the effects of an emergency and can provide basic services for approximately 24 h. Damage to some parts of the element/object can be expected in the event of a long-term emergency.	Good absorption
19–24	The element/object is characterized by a low level of absorption. In practice, this means that all material and physical resources are not sufficiently covered to withstand the effects of an emergency and to continue to provide basic services. Damage to individual elements can occur within approximately 3 h after the occurrence of an emergency.	Low absorption

**Table 5 ijerph-18-08286-t005:** Draft list of needs of the population.

List of Fictive Residents of the Municipality of Radôstka						
N.	Name	Surname	Address	Medical Equipment or Other Equipment Necessary for Household Life Connected to Electricity	Telephone Number	Solution Status
**1.**	Anna	Kudelčíková	Dlhá ulica č. 785	Home artificial lung ventilation	904 513 752	Primary
**2.**	František	Malý	Dlhá ulica č. 78	None	905 332 222	Secondary
**3.**	Adela	Strapatá	Dlhá ulica č. 985	Electric hydraulic patient lift	904 236 563	Primary
**x**	x	x	x	x	x	x

## Data Availability

All data used were correctly cited in the article with references to their sources. As part of the case study, some information was obtained through consultations in practice and was transformed into a written form. Some of the information used for security research of specific objects is sensitive. Information that is confidential or sensitive is not included in the article.

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
