# Peer review of "Methodological Framework for Resilience Assessment of Electricity Infrastructure in Conditions of Slovak Republic"

_ijerph, 2021, doi:10.3390/ijerph18168286_

Round 1

Reviewer 1 Report

In my opinion the paper is very much descriptive and contains no significant research to warrant publication. Instead the paper contains too much information on institutional details.

I might be wrong and hence an additional reviewer's opinion might be useful.

Author Response

Thank you for your opinion on the quality of the article. Yes, the language style used is rather descriptive, but it is based on the intention to give the general professional public possible guidance on how to increase the resilience of energy systems in order to minimize the impact on the health and lives of citizens. The ever-occurring crisis phenomena clearly show that it is not possible to assume that our systems are resilient on the basis of past technical standards and experience. The principles of continuous monitoring and improvement are key to the functionality of all, especially energy systems. The electricity subsystem is most important for maintaining the quality of life of society.

In the current version of the article, we have removed some irrelevant information that was not needed. Thanks for helping us improve our article.

Reviewer 2 Report

The paper has improved after the revision and now can be considered for publication. Please note that references in the abstract are uncommon and to be avoided.

Author Response

Thank you for your recommendation, we removed this part of abstract  and completly reworked abstract. 

Reviewer 3 Report

It addresses the comments on the previous version. 

Author Response

Thank you for your  recommendations, our paper we tried enhancing  by your comments ands recomendations.

This manuscript is a resubmission of an earlier submission. The following is a list of the peer review reports and author responses from that submission.

Round 1

Reviewer 1 Report

Below I submit my comments which I hope will help to improve the manuscript.

The abstract is very poorly written in terms of its informational contents. The sentence like “The researchers on University of Zilina have been creating theoretical framework in the field of resilience and critical infrastructure for 10 years.” sounds very awkward. The abstract should contain precise methodological contribution followed by important findings and conclusions.

In introduction the authors discuss 2001 attack, which I think has no direct connection to the Slovak power system. Instead integration processes of power infrastructure between Slovakia, Czech Republic, Romania and other countries should be discussed.

What is the relation of text in lines 76-78 to the current research? How is it related to public health area where the manuscript is submitted?

The English language used is low from the academic level. Expressions and phrases used in the manuscript are not quite standard.

Generally the paper is far from being publishable at this moment because it is too much descriptive bearing no significant scientific value added or quantitative analysis, which is usually a key for a publication.

Author Response

Response to Reviewer 1 Comments

Point 1:

The abstract is very poorly written in terms of its informational contents. The sentence like “The researchers on University of Zilina have been creating theoretical framework in the field of resilience and critical infrastructure for 10 years.” sounds very awkward. The abstract should contain precise methodological contribution followed by important findings and conclusions.

Response 1: Thank you for pointing out, abstract was written badly, we revised it completely.

Point 2:

In introduction the authors discuss 2001 attack, which I think has no direct connection to the Slovak power system. Instead integration processes of power infrastructure between Slovakia, Czech Republic, Romania and other countries should be discussed.

Response 2: Thank you for pointing out, reference to 2001 attack was inserted by mistake, we rewrote the paragraph.

Point 3:

What is the relation of text in lines 76-78 to the current research? How is it related to public health area where the manuscript is submitted?

Response 3: The article was written for Section Environmental Science and Egineering, Special Issue Managing Disaster Risk in a Changing World. The relation to the public health area is given by necessity of electricity supply for ensuring the basic life’s and peoples’ needs being ill and needing the support of electronic devices to keep up basic body functions.

Point 4:

The English language used is low from the academic level. Expressions and phrases used in the manuscript are not quite standard.

Response 4: Thank you for pointing out, English was significantly improved.

Point 5:

Generally the paper is far from being publishable at this moment because it is too much descriptive bearing no significant scientific value added or quantitative analysis, which is usually a key for a publication.

Response 5:  We eliminated descriptivness, we changed the style of writing and we strived to improve significant scientific value.

Reviewer 2 Report

This manscript gives a thorough review on relevant literature and backgound of the resilience electricity infrastructure in Slovak Republic. To many readers, it is an appropriate reading material to know the in-depth situation there. 

As a suggestion for improvement, the abstract needs to be reshaped. The current version does not articulate the purpose of this manuscript. The authors should directly tell the readers that the aim of the article was to propose a new methodological framework for increasing the resilience electricity infrastructure, and also give a compact but concrete explanation on how they achieve it. Some more details should be carefully selected and embedded into the abstract. 

Author Response

Response to Reviewer 2 Comments

Point 1:

This manuscript gives a thorough review on relevant literature and background of the resilience electricity infrastructure in Slovak Republic. To many readers, it is an appropriate reading material to know the in-depth situation there. 

Response 2: Thank you for commenting on.

Point 2:

As a suggestion for improvement, the abstract needs to be reshaped. The current version does not articulate the purpose of this manuscript. The authors should directly tell the readers that the aim of the article was to propose a new methodological framework for increasing the resilience electricity infrastructure, and also give a compact but concrete explanation on how they achieve it. Some more details should be carefully selected and embedded into the abstract. 

Response 2: Thank you for pointing out, whole abstract was reshaped into acceptable form.

Reviewer 3 Report

Resilience of electricity supply infrastructure is of utmost importance. The paper addresses this problem, but in view of the reviewer, the paper needs significant improvement before it can be published:

  • Abstract: the first 3 sentences fit to the introduction, but not abstract. In the abstract please focus on what the paper presents, but not why the paper is written. Avoid using plural 3rd person (they) in the abstract, use instead plural first person (we).
  • Abstract: ‘Within this framework, they concluded that electricity sector is the most significant in whole system of critical infrastructure.’ This not a new statement and no research is needed to derive such a statement.
  • Section 3. Literature review. The style of literature review is somewhat unclear. It seems you describe the paper in one paragraph and then provide citation at the end by a numbered reference [Nn] (Vancouver reference style). Please provide the citation as soon as you refer in the text to it. There is no need to copy the title and the authors unless you use Harvard referencing.
  • Citation: please list citations in the order they are cited – check the numbering.
  • Figure 1: please provide resilience definition not for farming system, but for energy system or at least engineering domain. The example is very informative for this paper
  • Section 4: Research methods. It is not clear which research methods were used and how. The section is very abstract and provides almost no useful information.
  • Section 5: Results.
    • The organisation of the section is not optimal for a research paper. If you provide information on the research projects, please use the literature review section for that (section 5.1).
    • What are ‘Real results’? (section 5.2)
    • The case study should be provided in a separate section, not within the ‘Results’
    • The authors state that Figure 2 provides ‘a new, unique, and comprehensive framework for resilience’. In fact it is a duplication of a risk assessment process that is well known and standardised. Please discuss the link between risk assessment and resilience assessment.
    • The case study seems to focus on floods only, but this is only one hazard. If you work on a framework, it is important to describe how you treat other hazards, threats and their interactions. This part of the paper must be improved.

Author Response

Response to Reviewer 3 Comments

Point 1:

Resilience of electricity supply infrastructure is of utmost importance. The paper addresses this problem, but in view of the reviewer, the paper needs significant improvement before it can be published:

  • Abstract: the first 3 sentences fit to the introduction, but not abstract. In the abstract please focus on what the paper presents, but not why the paper is written. Avoid using plural 3rdperson (they) in the abstract, use instead plural first person (we).
  • Abstract: ‘Within this framework, they concluded that electricity sector is the most significant in whole system of critical infrastructure.’ This not a new statement and no research is needed to derive such a statement.“

Response 1: Thank you for pointing out, abstract was thoroughly revised.

 Point 2:

  • Section 3. Literature review. The style of literature review is somewhat unclear. It seems you describe the paper in one paragraph and then provide citation at the end by a numbered reference [Nn] (Vancouver reference style). Please provide the citation as soon as you refer in the text to it. There is no need to copy the title and the authors unless you use Harvard referencing.

Response 2: Thank you, literature review was improved, various reference styles were unified.

Point 3:

  • Citation: please list citations in the order they are cited – check the numbering.
  • Figure 1: please provide resilience definition not for farming system, but for energy system or at least engineering domain. The example is very informative for this paper

Response 3: Thank you for pointing out, the list of references was fixed, the list citation was improved.

Point 4:

  • Section 4: Research methods. It is not clear which research methods were used and how. The section is very abstract and provides almost no useful information.

Response 4: The research methods were completely revised. There are used only those methods that were used in fact now.

Point 5:

  • Section 5: Results.
    • The organization of the section is not optimal for a research paper. If you provide information on the research projects, please use the literature review section for that (section 5.1).
    • What are ‘Real results’? (section 5.2)
    • The case study should be provided in a separate section, not within the ‘Results’
    • The authors state that Figure 2 provides ‘a new, unique, and comprehensive framework for resilience’. In fact it is a duplication of a risk assessment process that is well known and standardised. Please discuss the link between risk assessment and resilience assessment.
    • The case study seems to focus on floods only, but this is only one hazard. If you work on a framework, it is important to describe how you treat other hazards, threats and their interactions. This part of the paper must be improved.

Response 5:  Section 5: Results was significantly improved. Case study was replaced into sole section. Focus on the flood was explained. The main aim of the article is to see its uniqueness within the context of Slovak republic. Moreover it was consulted with the scholars in practice. Thank you for suggestions at improving of the article.

Round 2

Reviewer 1 Report

I appreciate improvements in the abstract. Let me provide comments for further possible improvements of the abstract. Based on these comments, it would be very important that authors improve similarly the rest of the text.

I think that the first two sentences in lines 10-13 should be deleted.

Replace “the scholars adopt different approach” in lines 23-24 by “various approaches have been adopted”

Delete “and has retained so far” in line 21.

Replace “Specifically, writers of the paper” in line 25 by “We”

Information in the sentence in lines 27-29 is not necessary.

Replace “done onto the issue showed” in line 29 by “shows”

Replace “methodical” in line 33 by “methodological”

The major defect in this manuscript is that it provides too much of secondary unimportant information. The research article should be rather more focused, without too many extra distracting details. A particular example is in lines 83-96. Why the authors talk about PhD students, testing and vaccine development.

Similarly, I think some of information should be more concise in line 50-68.

Does information in lines 125-154 really represent the state of the art? I doubt. Probably this section is more about Background.

There are still too many English language mistakes, which should have been corrected before submission. For example, “the Slovakia” in line 167, “on Slovakia” in line 170, “4081 Megawatts of installed power” in line 170 should be rather “4081 Megawatts of installed capacity”

Please rewrite the paper accordingly so that a reader is able to understand value-added of this research. Otherwise, with so much secondary and sometimes irrelevant information it is not possible to grasp the value-added of this research, if there is any scientific value-added at all.

Reviewer 3 Report

I looked at the responses the Authors have provided. They claim they have revised most of the manuscript and the highlighted areas. However, when looking at the text it is not so evident.

The manuscript is still in draft mode - check numbering of sections 4 and 5, some headings are still in Slovak and not in English.